# Homotypic interactions between SLAMF1 receptors on innate T cells and neutrophils regulate killing of fungi

Lindsay Suarez Lau[1], Sarah Lichtenberger[1], Cleison Ledesma Taira[1], Bruce S. Klein[1,2,3], Marcel Wüthrich [1]*

1 Department of Pediatrics, University of Wisconsin, Madison, Wisconsin, United States of America, 2 Department of Internal Medicine, University of Wisconsin, Madison, Wisconsin, United States of America, 3 Department of Medical Microbiology and Immunology, University of Wisconsin, Madison, Wisconsin, United States of America

* mwuethri@wisc.edu

## Abstract

Innate lymphocytes and myeloid cells communicate and play an essential part in activating neutrophils and other effector cells to kill fungi. Here, we identified that Signaling Lymphocytic Activation Molecule 1 (SLAMF1) orchestrates a cellular and molecular signaling network that activates phagocytes. We uncovered innate lymphocytes including innate CD4+ or TCRγδ+ T cells augment neutrophil killing of *Blastomyces dermatitidis* (*Bd*) in a SLAMF1 dependent manner. SLAMF1 expression on neutrophils enabled homotypic SLAMF1:SLAMF1 interactions with innate CD4+ T cells, which released soluble factors that activated neutrophils to kill fungi. Our work furnishes new mechanistic insight about the role of SLAMF1 in mobilizing innate immune cells to induce phagocyte-driven killing of inhaled fungi.

### Author summary

The innate immune system is the first line of defense against inhaled fungi but the mechanisms by which the host immune system becomes activated and mounts a protective response is incompletely understood. We identified a receptor on innate immune cells that facilitates communication between cells and recruits and activates killer cells that destroy inhaled fungi. We found that a receptor on the cell surface of innate immune cells promotes cell contact and release of soluble factors, which orchestrates resistance to infection. We envision that soluble receptor could be harnessed for future anti-fungal therapy.

**Data availability statement:** All relevant data are within the manuscript, its Supporting Information files.

**Funding:** NIH grants R21AI173718 (MW), R01AI040996 (BK/MW), R37 AI035681 (BK), and R01 AI168370 (BK) The funders had no role in study design, data collection and analysis, decision to publish, or preparation of the manuscript.

**Competing interests:** The authors have declared that no competing interests exist.

## Introduction

Cell surface molecules expressed on leukocytes regulate innate immune responses to pathogens. Signaling Lymphocytic Activation Molecule Family (SLAMF) receptors orchestrate and interconnect innate and adaptive immune cells [1]. SLAMF receptors are expressed on hematopoietic lineage cells [2] and have three modes of action: (i) homophilic contact-dependent interaction between cells that modulate immunity; (ii) direct microbial sensing as part of the anti-microbial response; and (iii) as an entry receptor for pathogens [3–5]. The role of SLAMF1 in antifungal resistance is understudied. We have reported that SLAMF1 is dispensable in vaccine immunity [6], but essential for host defense against fungi. Thus, understanding its role in innate immunity represents a gap in knowledge. Herein, we investigated whether hematopoietic cells activate phagocytes to kill fungal pathogens through homophilic signaling between SLAMF1 receptors on innate lymphocytes and myeloid cells.

## Results & discussion

### SLAMF1 is required for innate resistance to Bd

To investigate whether innate resistance to *Bd* infection requires SLAMF1, we infected unvaccinated mice. SLAMF1$^{-/-}$ mice died earlier (**Fig 1A**) and had 3.5 log more lung CFU than wild-type mice at the termination of the experiment (**Fig 1B**), indicating a deficit in innate resistance to fungal infection in SLAMF1$^{-/-}$ mice.

Phagocytes including neutrophils, monocytes, and macrophages internalize and kill fungi [7]. To see if SLAMF1 impacts elimination of *Bd* by phagocytes *in vivo*, we used a DsRed viability reporter strain [8,9]. We assessed killing by phagocytes at 16 hours post-infection (hpi) when the burden of lung CFU was comparable between wild type and SLAMF1$^{-/-}$ mice. The percentage of dead yeast in neutrophils and monocytes, but not alveolar macrophages, was reduced in SLAMF1$^{-/-}$ mice vs. controls (**Figs 1C** and **S1A**). Thus, killing of *Bd* by neutrophils and monocytes is impaired in the absence of SLAMF1 [7]. While the percentage of killed yeast associated with neutrophils (26.9%) is lower than with monocytes (49.08%), the number of yeast-associated neutrophils is 4.3 times higher than yeast-associated monocytes (**S1B Fig**) indicating that innate resistance is mostly determined by killing of yeast by neutrophils. We thus focused on activation of neutrophils to understand how SLAMF1 promotes resistance.

Production of reactive oxygen species (ROS) and nitric oxide (NO) is essential and serves as a surrogate of innate antifungal resistance [8]. We investigated a link between these products and reduced *in vivo* killing by neutrophils in SLAMF1$^{-/-}$ mice. ROS and NO staining were reduced in yeast-associated neutrophils and monocytes in SLAMF1$^{-/-}$ vs. controls (**Fig 1D**) and was unaffected in alveolar macrophages.

To assess intrinsic killing by SLAMF1$^{-/-}$ neutrophils and monocytes, we cocultured the cells with *Bd in vitro*. In contrast to the *in vivo* defect, *in vitro* killing of *Bd* was similar for SLAMF1$^{-/-}$ and SLAMF1$^{+/+}$ neutrophils and monocytes. (**Fig 1E** and **1F**). Thus, these SLAMF1$^{-/-}$ phagocytes are not intrinsically impaired in killing *Bd* yeast. SLAMF1 must be required on other cells for antifungal function of neutrophils and monocytes *in vivo*.

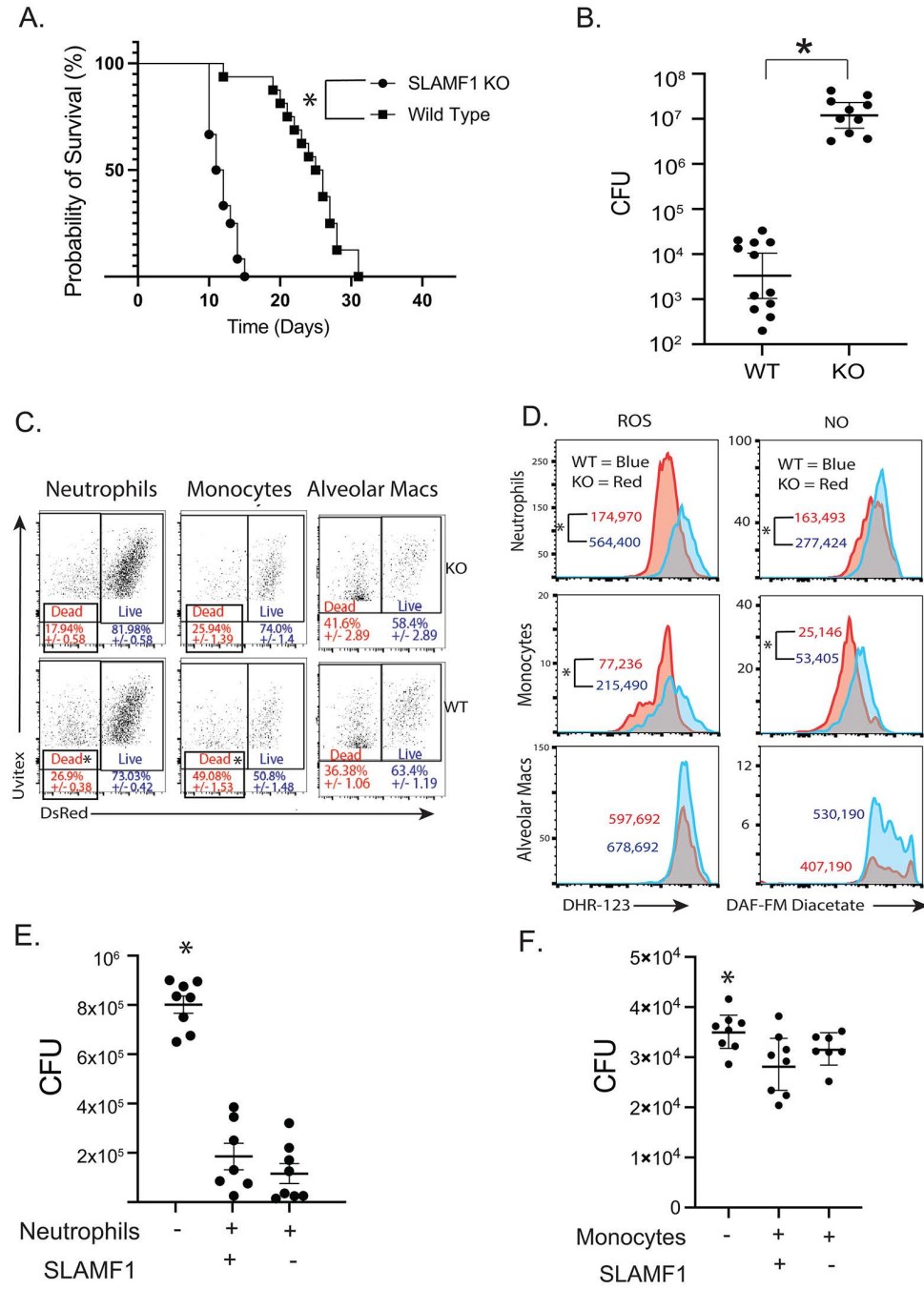

**Fig 1. SLAMF1 is required for innate resistance to *Bd*, *in vivo* killing and ROS/NO production by neutrophils and monocytes.** Survival **(A)** and lung CFU **(B)** 4 days post infection of SLAMF1 knockout (KO) and wild type (WT) mice infected with *Bd*. *$p < 0.05$, Kaplan Meier test for survival. CFU from at least 10 mice/group are expressed as $\text{Log}_{10}$ plotted with geometric mean ± geometric SD *$p < 0.05$, two tailed Mann-Whitney T test. *In vivo* killing assay: Dot plots show yeast-associated cells (alveolar macrophages, neutrophils and monocytes) and the percentages of cells containing dead (DsRed-, Uvitex+) vs. live (DsRed+, Uvitex+) yeast. Dot-plots are concatenated data for 5 mice/group *$p < 0.05$ vs KO, two tailed Mann-Whitney T test **(C)**. Lung cells stained *ex vivo* with ROS or NO indicators DHR-123 or DAF-FM diacetate. Geometric MFI of staining is shown for yeast-associated neutrophils, monocytes, and alveolar macrophages *$p < 0.05$ two tailed Mann-Whitney T test **(D)**. *In vitro* killing assay: neutrophils and monocytes were isolated from bone marrow of wildtype and SLAMF1 KO mice and cocultured with *Bd* yeast overnight at 37C. 8 replicates plated for CFU. *$p < 0.05$ vs. all other groups, two tailed Mann-Whitney T test **(E + F)**.

### Innate CD4+ T cells, TCRγδ⁺ T cells and monocytes express SLAMF1 and augment neutrophil killing of Bd in vitro

We stained lung cells for SLAMF1 16 hpi when lung CFU is similar between SLAMF1$^{-/-}$ and control mice. Among 17 populations of myeloid and lymphoid cells surveyed, CD4$^+$TCRβ$^+$, TCRγδ$^+$, MAIT cells and Ly6C$^{hi}$ CCR2$^+$ monocytes expressed SLAMF1 (**Fig 2A**). We next investigated whether these cells activate neutrophils in a SLAMF1-dependent manner. MAIT cells were excluded, as there were too few in the lung after infection and specific cre mice were unavailable to study the SLAMF1 on MAIT cells.

We established an *in vitro* coculture assay with yeast, neutrophils and each candidate cell type (**Fig 2B**). We used naïve neutrophils from bone marrow and SLAMF1$^+$ candidate cells from lungs 16 hpi; cell purity was > 90% for CD4$^+$ T cells and monocytes, and 56% for TCRγδ$^+$ T cells (**S2 Fig**). Whereas neutrophils alone reduced yeast numbers (**Fig 2C - 2E**), the addition of CD4$^+$ T cells, TCR-γδ$^+$ T cells or monocytes to cocultures further reduced CFU (**Fig 2C -2E**). Thus, CD4$^+$ T cells, TCRγδ$^+$ T cells and monocytes augment neutrophil killing of fungi. CD4$^+$ T cells also augmented fungal killing by monocytes (**Fig 2F**).

### Conditional mice that lack SLAMF1 on innate lymphocytes or monocytes

We created mice where SLAMF1 is ablated on the target cell by Tamoxifen treatment. We validated commercial SLAMF1$^{fl/fl}$ mice by creating EIIa cre x SLAMF1$^{fl/fl}$ mice that lack SLAMF1 in embryonic cells. They phenocopy SLAMF1 cell surface staining and resistance to *Bd* infection of SLAMF1 global knockout mice (**S3A3-S3D Fig**).

To test the role of SLAMF1 on target cells, we crossed SLAMF1$^{fl/fl}$ mice with cell-specific Cre$^+$ mice (**S3E3-S3H Fig**). We verified loss of SLAMF1 on target cells by FACS. At 16 hpi, CD4$^+$ T cells from the lungs of CD4-Cre$^+$ x SLAMF1$^{fl/fl}$ mice showed reduced SLAMF1 staining vs. controls (**S3E Fig**). Likewise, SLAMF1 was reduced on TCRγδ$^+$ T cells from TCRγδ-Cre$^+$ x SLAMF1$^{fl/fl}$ mice (**S3F Fig**) and monocytes from CCR2-Cre$^+$ x SLAMF1$^{fl/fl}$ mice (**S3G Fig**). Since wild type monocytes display limited SLAMF1, we used GFP as a surrogate marker to verify Cre activation and elimination of SLAMF1 after tamoxifen treatment on the cells from CCR2-Cre$^+$ x SLAMF1$^{fl/fl}$ mice (**S3H Fig**), but not controls.

### CD4+ T cells and TCRγδ+ T cells require SLAMF1 to augment killing by neutrophils

To investigate whether CD4$^+$ T cells, TCRγδ$^+$ T cells and monocytes require SLAMF1 to augment neutrophil killing, we cocultured yeast and neutrophils with candidate cells from Cre$^+$ and Cre$^-$ conditional mice (**Fig 3A**). CD4$^+$ T cells that lack SLAMF1 in CD4-Cre$^+$ mice failed to augment neutrophil killing compared to CD4-Cre- controls (**Fig 3B**). Likewise, TCRγδ$^+$ T cells from TCRγδ-Cre$^+$ mice failed to augment neutrophil killing compared to Cre- controls (**Fig 3C**). Monocytes from CCR2-Cre$^+$ mice did not differ from Cre$^-$ controls in this assay (**Fig 3D**). Thus, only innate T-cells augment killing by neutrophils in a SLAMF1-dependent manner.

### SLAMF1 on CD4+ T cells is indispensable for anti-fungal resistance in vivo

We challenged conditional knockout mice and analyzed lung CFU to determine dispensability of SLAMF1 on target cells. Similar to SLAMF1 global knockout mice, CD4-Cre$^+$ mice that lacked SLAMF1 on CD4$^+$ T cells had increased lung CFU vs. controls (**Fig 3E**). In contrast, we saw no difference in lung CFU between TCRγδ-Cre$^+$ vs. Cre$^-$ mice and CCR2-Cre$^+$ vs. Cre$^-$ mice (**Fig 3E**). Thus, SLAMF1 is indispensable on CD4$^+$ T cells for host resistance but dispensable on TCRγδ$^+$ T cells and monocytes.

### Augmented neutrophil killing also requires SLAMF1 expression on neutrophils

CD4$^+$ T cells and TCRγδ$^+$ T cells required SLAMF1 to augment neutrophil killing in vitro. If these lymphocytes activate neutrophils by SLAMF1:SLAMF1 homotypic interactions, SLAM1 expression should be required on neutrophils. To test this, we cocultured CD4$^+$ T cells and neutrophils from wild type or SLAMF1$^{-/-}$ mice. Neutrophils alone from wild type or SLAMF1$^{-/-}$ mice each reduced yeast numbers, confirming SLAMF1$^{-/-}$ neutrophils have no intrinsic killing defect. The addition of CD4$^+$ T

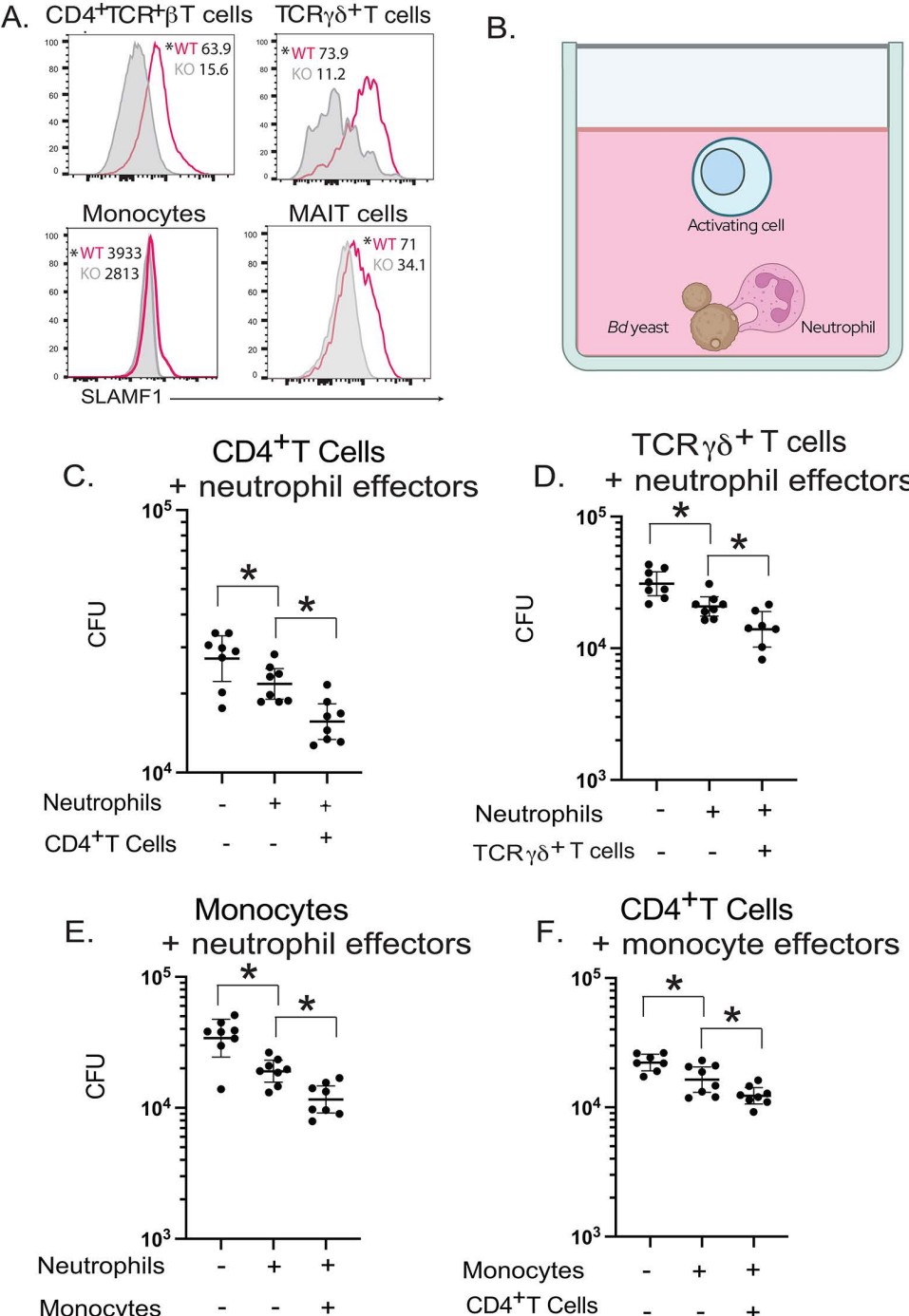

**Fig 2. CD4⁺ T cells, TCRγδ⁺ T cells, and monocytes express SLAMF1 and augment phagocyte killing of *Bd*.** *Ex vivo* staining of lung leukocytes for SLAMF1. WT and SLAMF1 KO mice were challenged with *Bd* and lungs harvested at 16 hr post-infection. Lung cells were stained *ex vivo* with anti-SLAMF1 (CD150) mAb. SLAMF1 staining of cells from WT and SLAMF1 KO mice (shaded. The plots are concatenates from 5 mice/group. P values were calculated based on individual sample MFIs between WT and SLAMF1KO mice. p < 0.05 vs. KO for all cell types, two tailed Mann-Whitney T test **(A)**. *In vitro* killing assay with candidate activating cells, neutrophils and *Bd* yeast **(B)** created in Biorender https://BioRender.com/lc170nt. Neutrophils were isolated from bone marrow of WT mice. Candidate cells were isolated from lungs of WT mice 16 hr after challenge. Neutrophils plus CD4⁺ T cells **(C)**, TCRγδ⁺ cells **(D)** and monocytes **(E)** were cocultured with *Bd* yeast overnight. In **(F)**, CD4⁺ T cells were co-cultured with *Bd* yeast and monocytes as the effector. 8 replicates plated for CFU. *p < 0.05 two tailed Mann-Whitney T test. Bars are geometric mean with 95% CI. Graphs representative of 5 **(B)**, 2 **(C)**, and 2 **(D)** experiments, respectively.

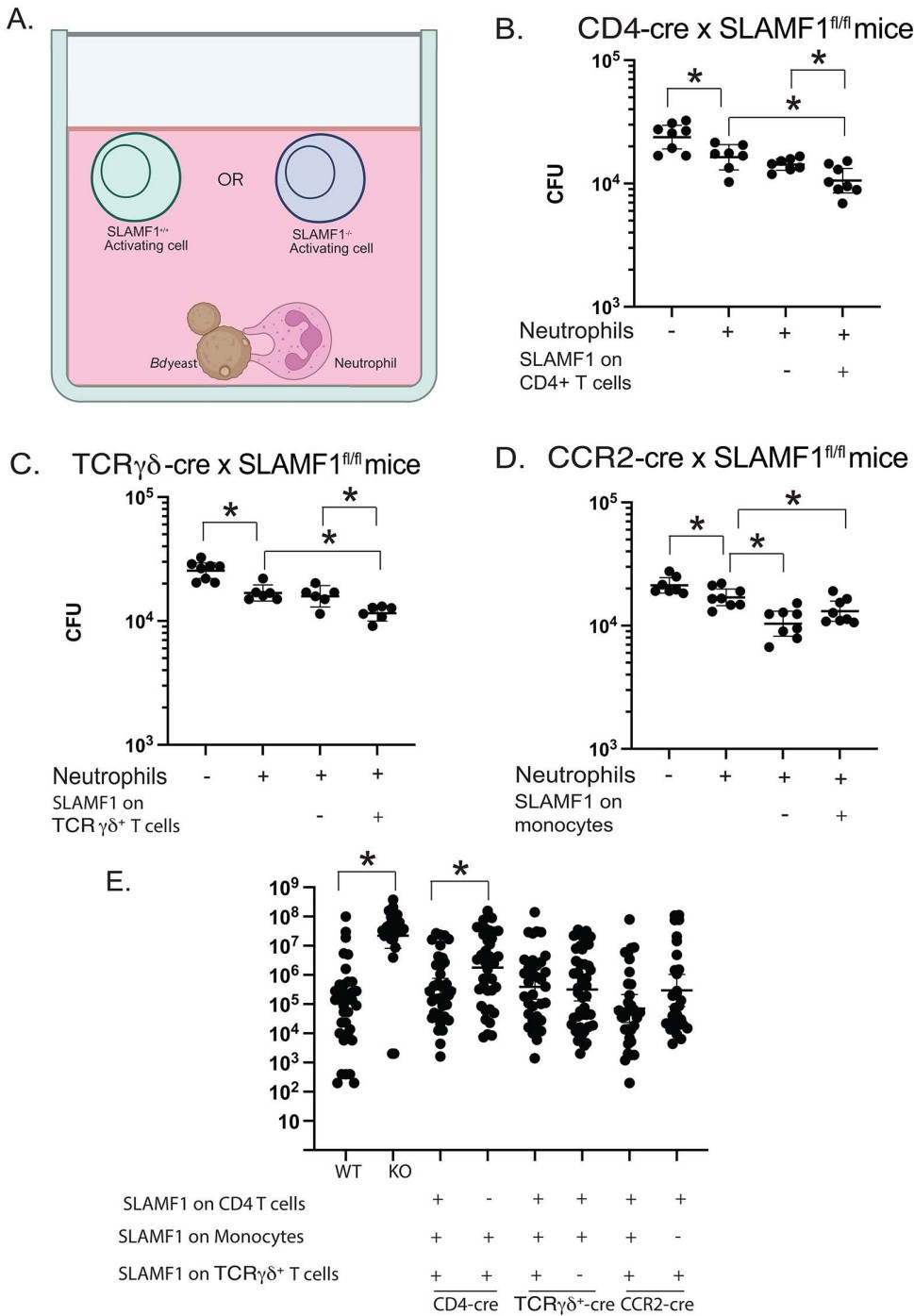

**Fig 3. CD4⁺ T cells and TCRγδ⁺ T cells augment neutrophil killing in a SLAMF1-dependent manner and *in vivo* killing of *Bd* in conditional SLAMF1knockout mice.** *In vitro* killing assay with candidate activating cells from conditional SLAMF1 knockout mice **(A)** created in Biorender https://BioRender.com/71w6m5v. Neutrophils isolated from bone marrow of wild type mice were cocultured with *Bd* yeast and candidate activating cells from lungs of conditional CD4-Cre x SLAMF1fl/fl **(B)**, TCRγδ-Cre x SLAMF fl/fl mice **(C)**, and CCR2-cre SLAMF1fl/fl mice **(D)** harvested 16 hr post-infection. 8 replicates plated for CFU. Data representative of 3 assays. *p<0.05, two tailed Mann-Whitney T test. Wild type (WT), SLAMF1 knockout (KO), conditional CD4-cre x SLAMF1fl/fl mice, CCR2-cre x SLAMF1fl/fl mice, and TCRγδ-cre x SLAMF1fl/fl mice were infected with 2x10⁴ *Bd* yeast and lung CFU plated at 14 days post-infection. Cre⁺ mice lack SLAMF1 on corresponding conditional knockout mice, Cre⁻ mice served as wildtype littermate controls for the corresponding Cre⁺ knockout mice **(E)**. *p<0.05 two tailed Mann-Whitney T test. Bars represent geometric mean with 95% CI.

cells from wild type mice reduced CFU in the wells with neutrophils from wild type but not SLAMF1$^{-/-}$ mice (**Fig 4A**). Thus, homotypic SLAMF1 interaction between neutrophils and CD4$^+$ T cells is required to augment killing by neutrophils.

Unexpectedly, we saw no SLAMF1 expression on lung neutrophils from infected wild type mice (**S4A Fig**) despite *in vitro* results indicating SLAMF1 on neutrophils is required to augment killing. We surmised SLAMF1 on neutrophils could be internalized upon contact with *Bd* yeast in the lung, as the *in vitro* assay uses naïve neutrophils from bone marrow. Indeed, fresh neutrophils from the bone marrow of wild type mice display SLAMF1 (**S4B Fig**). We then tested if SLAMF1 on neutrophils is downregulated by exposure to *Bd* yeast *in vitro*. Neutrophils from wild type mice displayed SLAMF1 when cultured without yeast but not after coculture with yeast (**S4C Fig**). After coculture, we detected intracellular SLAMF1, suggesting the receptor had become internalized. Thus, early in infection, homotypic SLAMF1:SLAMF1 interactions between CD4$^+$ T cells and neutrophils likely augment neutrophil killing of yeast. Yeast may subvert these events by binding the receptor themselves.

*Mtb* infection conversely enhances expression of SLAMF1 on human neutrophils [10]. That study concluded that SLAMF1 can regulate myeloid cell autophagy or Th1 responses against *Mtb*. Hence, SLAMF1 may offer a target for host directed therapy by increasing pathogen control or reducing tissue damage during infection.

### Homotypic SLAMF1 interactions between innate CD4+ T cells and neutrophils augment fungal killing by neutrophils via production of soluble factors

We investigated whether homotypic SLAMF1 interactions between innate T cells and neutrophils yields soluble factors that activate neutrophils to kill *Bd*. We established a transwell assay where the lower well contained yeast, innate T cells and neutrophils, and the upper well contained only neutrophils and yeast. Soluble factors from the lower well could then pass the membrane to activate neutrophils in the upper well (**Fig 4B**). To verify that soluble factors pass through the 0.4 mm pores of the transmembrane, we used recombinant IFN-γ (17 kDA) to activate neutrophils and increase their release of ROS and pro inflammatory cytokines [11]. IFN-γ addition to the lower well augmented neutrophil killing of *Bd* in the top well (**S4D Fig**). Addition of CD4$^+$ T cells to the bottom well augmented killing of yeast by neutrophils in both the bottom well and top well (**Fig 4C**). Thus, homotypic SLAMF1 interactions between CD4$^+$ T cells and neutrophils yield soluble factors that also activates neutrophils. We conclude that augmented killing by neutrophils may result from SLAMF-1 mediated cell-contact or contact-dependent production of soluble factors or both (**Fig 4D**).

We previously reported that TCRγδ$^+$ T cells and CD4$^+$ T cells recruit and activate neutrophils and innate resistance to *Bd* infection [12]. We did not investigate the role of SLAMF1 in that study. The mechanics that activate these innate populations at the site of infection have not been fully identified to our knowledge. Homotypic SLAMF1 interactions between innate lymphocytes and neutrophils and associated release of soluble products may underpin such responses during *Bd* and other mucosal fungal infections (e.g., *candidiasis)*, where neutrophil killing of fungi could result from SLAMF1 mediated cell-contact, contact-dependent production of soluble factors or both.

Our study raises questions about SLAMF1 that have implications for innate immunity to fungi and other intracellular microbes: SLAMF1 as a sensor of fungi and the receptor's role in either phagocytosis or immune evasion; SLAMF1 induced signaling in lymphocytes and neutrophils; the identity of soluble factors released by SLAMF1 homotypic interactions; and the potential use of soluble SLAMF1 receptor for antifungal immunotherapy. Insights from such work will deepen understanding of diverse SLAMF1's receptor functions that regulate innate immune interactions and provide the foundation for targeted host directed therapies.

## Materials and methods

### Ethics Statement

Animal studies adhered to protocol M005891 approved by the IACUC of UW-Madison. Animal studies were compliant with provisions established by the Animal Welfare Act and the Public Health Services (PHS) Policy on the Humane Care and Use of Laboratory Animals.

Additional Methods are found in S1 Text.

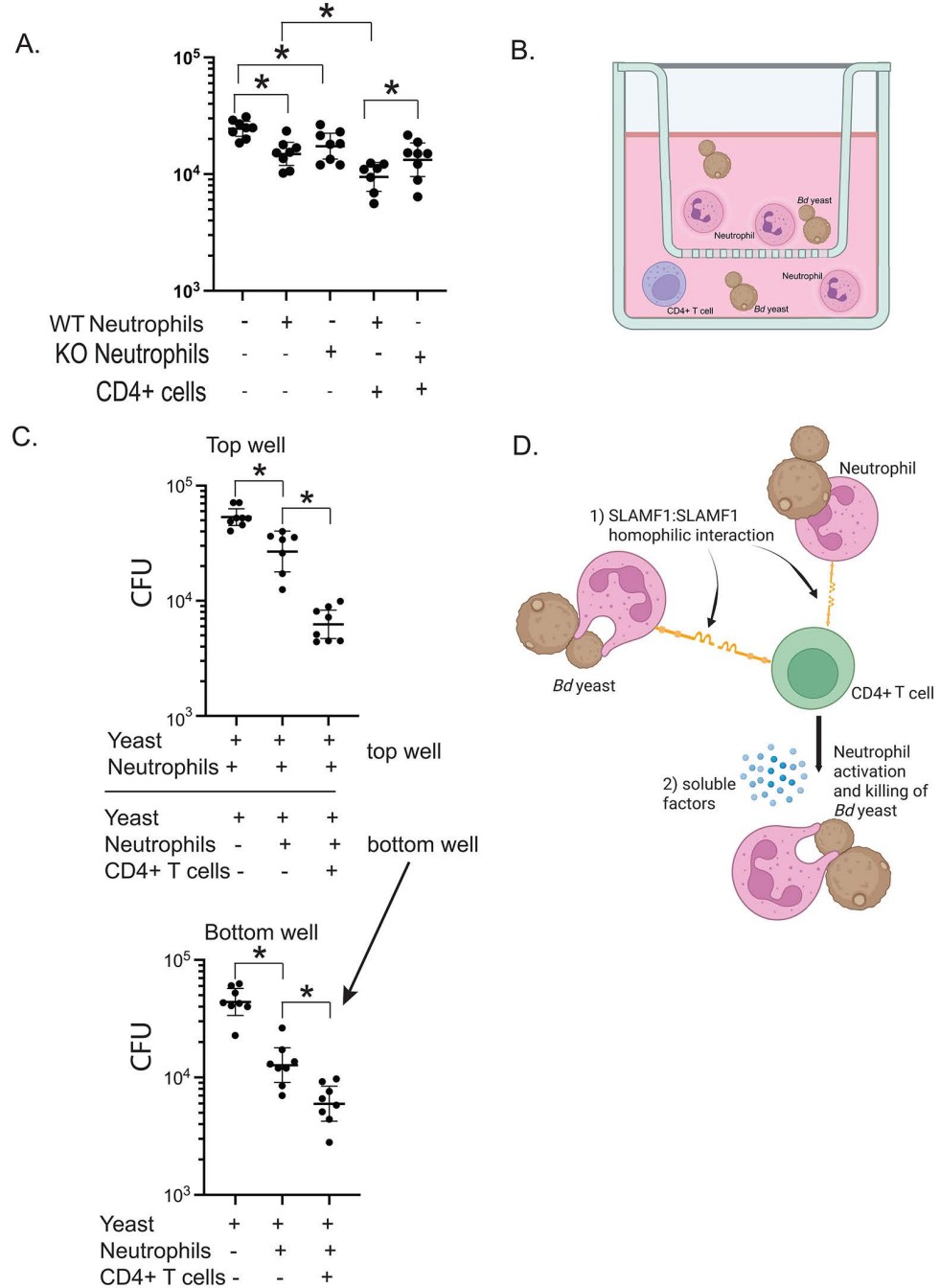

**Fig 4. Augmented neutrophil killing requires SLAMF1 expression on neutrophils and soluble factors.** *In vitro* killing assay with wild type or SLAMF1 KO neutrophils and CD4+ T cells from WT mice. Neutrophils from bone marrow of naive mice were cocultured overnight with *Bd* yeast and CD4+ T cells harvested from the lungs of WT mice 16 hrs post-infection **(A)**. 8 replicates plated for CFU. *p < 0.05, two tailed Mann-Whitney T test. Format of *in vitro* transwell assay: activating cells that potentially produce soluble factor in the bottom well; sensing neutrophils in the top well. *Bd* yeast and neutrophils from bone marrow of naive wild type mice were placed in the top insert wells. *Bd* yeast, neutrophils, and CD4+ T cells harvested from the lungs of wild type mice 16 hr post challenge were placed in the bottom well **(B)** created in Biorender https://BioRender.com/lc170nt. Transwell system results after overnight coculture, 8 replicates plated for CFU **(C)**. *p < 0.05, two tailed Mann-Whitney T test. Model for SLAMF1 functions **(D)** created in Biorender https://BioRender.com/2r3dje9: 1) Cell contact through SLAMF1: SLAMF1 homotypic interactions between neutrophils and CD4+ T cells are requisite for neutrophil activation. 2) Homotypic SLAMF1:SLAMF1 interactions generate soluble factors that also augment neutrophil killing of *Bd* yeast.

## Supporting information

**S1 Fig. To main Fig 1 *In vivo* killing assay: Percent of lung phagocytes associated with dead yeast (A) and number of alveolar macrophages, neutrophils, and monocytes associated with total number of yeast for wild type and SLAMF1 knockout mice (B).** $*p < 0.05$, two tailed Mann-Whitney T test.
(EPS)

**S2 Fig. To main Fig 2 Flow analysis and stacked bar graphs of lung leukocyte populations before and after isolation of CD4+ T cells (A) and TCRγδ+ T cells (B), and analysis of bone marrow before and after isolation of monocytes (C).** To identify cell types of interest before and after enrichment we used a panel of multiplexing antibodies to identify 17 pulmonary leukocyte subsets with 12 color flow cytometry [13]. Stacked bar graphs show percentage of each immune cell type relative to all immune cells identified by the Lymphoid Myeloid panel.
(EPS)

**S3 Fig. To main Fig 3 Breeding scheme for the generation of conditional SLAMF1knockout mice.** To validate commercial SLAMF1 floxed mice we bred them first to homozygosity and crossed them with Ella-Cre mice to generate Ella-cre x SLAMF1fl/fl mice that lack SLAMF1 in embryonic cells (A) created in Biorender https://BioRender.com/1ni5v4x. SLAMF1 expression on target cells for Ella-cre x SLAMF1fl/fl mice in comparison to wildtype and SLAMF1 KO mice. Geometric MFI of staining. Plots are concatenates from 5 mice/group. $*p < 0.05$ vs WT, two tailed Mann-Whitney T test (B). Survival (C) and lung CFU (D) 9 days post infection of Ella-cre x SLAMF1fl/fl mice and WT mice infected with *Bd*. $*p < 0.05$, Kaplan Meier test for survival. CFU from at least 9 mice/group are expressed as $Log_{10}$ plotted with geometric mean ± geometric SD $*p < 0.05$ two tailed Mann-Whitney T test. *Ex vivo* staining of lung leukocytes for SLAMF1 in CD4-cre x SLAMF1fl/fl mice (E) and TCRγδ-cre x SLAMF1fl/fl mice (F). *Ex vivo* staining of bone marrow for SLAMF1 in CCR2-cre x SLAMF1fl/fl mice (G). Plots show the histograms and geometric mean fluorescent intensity (MFI) of SLAMF1 staining for corresponding target cells in Cre+ and Cre- mice in comparison to wild type and SLAMF1 knockout mice. GFP expression by conditional CCR2-Cre+ x SLAMF1fl/fl mice that were generated using *CCR2-CreER-GFP* mice (H). Plots are concatenates from 5 or more mice/group. $*p < 0.05$ vs SLAMF1 expressing cells (WT vs KO, Cre- vs Cre+), two tailed Mann-Whitney T test.
(EPS)

**S4 Fig. To main Fig 4 *Ex vivo* staining of neutrophils from lungs 16 hr post-infection (A) and from bone marrow of naive mice (B).** Surface and intracellular staining of neutrophils from the bone marrow with anti-SLAMF1 (CD150) after overnight coculture with or without *Bd* yeast (C) Geometric MFI of staining. Plots are concatenates from 5 mice/group. Validation of transwell system using IFN-γ as a positive control (D). Neutrophils and yeast were placed in the top well and recombinant IFN-γ added in the bottom well. 16 hours later, CFU were plated from the top well.
(EPS)

**S1 Text. Materials and Methods.**
(DOCX)

**S1 Data. Raw data used to generate graphs.**
(XLSX)

## Acknowledgments

Flow samples were processed at the University of Wisconsin Carbone Cancer Center (UWCCC) Flow Core Facility on a BD LSR Fortessa and Aurora that was purchased with the NIH shared instrumentation grant 1S10OD018202–01 and University of Wisconsin Carbone Cancer Center Support grant P30 CA014520.

## Author contributions

**Conceptualization:** Bruce S Klein, Marcel Wüthrich.

**Data curation:** Lindsay Suarez Lau, Sarah Lichtenberger, Cleison Ledesma Taira, Marcel Wüthrich.

**Formal analysis:** Lindsay Suarez Lau, Sarah Lichtenberger, Cleison Ledesma Taira, Marcel Wüthrich.

**Funding acquisition:** Bruce S Klein, Marcel Wüthrich.

**Investigation:** Lindsay Suarez Lau, Marcel Wüthrich.

**Project administration:** Marcel Wüthrich.

**Supervision:** Marcel Wüthrich.

**Validation:** Lindsay Suarez Lau, Sarah Lichtenberger.

**Visualization:** Lindsay Suarez Lau, Sarah Lichtenberger, Cleison Ledesma Taira.

**Writing – original draft:** Lindsay Suarez Lau.

**Writing – review & editing:** Bruce S Klein, Marcel Wüthrich.

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
