## [Decision Letter · Decision Letter 0]

26 Mar 2026

PPATHOGENS-D-26-00450

Homotypic interactions between SLAMF1 receptors on innate T cells and neutrophils regulate killing of fungi

PLOS Pathogens

Dear Dr. Wuthrich,

Thank you for submitting your manuscript to PLOS Pathogens. After careful consideration, we feel that it has merit but does not fully meet PLOS Pathogens's publication criteria as it currently stands. Therefore, we invite you to submit a revised version of the manuscript that addresses the points raised during the review process.

We look forward to receiving your revised manuscript.

Kind regards,

Amariliz Rivera

Academic Editor

PLOS Pathogens

Michal Olszewski

Section Editor

PLOS Pathogens

Sumita Bhaduri-McIntosh

Editor-in-Chief

PLOS Pathogens

orcid.org/0000-0003-2946-9497

Michael Malim

Editor-in-Chief

PLOS Pathogens

orcid.org/0000-0002-7699-2064

**Additional Editor Comments :**

The current study provides convincing data to support the conclusions presented. The findings are of interest to a broad audience and will open new areas for investigation. Please address the reviewer's recommendations to modify the text.

**Journal Requirements:**

1) Please insert an Ethics Statement at the beginning of your Methods section in the main file of the manuscript, under a subheading 'Ethics Statement'. It must include:

i) The full name(s) of the Institutional Review Board(s) or Ethics Committee(s)

ii) The approval number(s), or a statement that approval was granted by the named board(s).

3) We have noticed that you have uploaded Supporting Information files, but you have not included a complete list of legends. Please add a full list of legends for your Supporting Information file (S1 Text.pdf) after the references list.

4) We note that your Data Availability Statement is currently as follows: "All relevant data are within the manuscript and its Supporting Information files.". Please confirm at this time whether or not your submission contains all raw data required to replicate the results of your study. Authors must share the “minimal data set” for their submission. PLOS defines the minimal data set to consist of the data required to replicate all study findings reported in the article, as well as related metadata and methods (https://journals.plos.org/plosone/s/data-availability#loc-minimal-data-set-definition).

1) State what role the funders took in the study. If the funders had no role in your study, please state: "The funders had no role in study design, data collection and analysis, decision to publish, or preparation of the manuscript.".

**Reviewers' Comments:**

Reviewer's Responses to Questions

**Part I - Summary**

Reviewer #1: In this report, Lau et al. describe how SLAMF1 governs homotypic interactions between CD4+ T cells and neutrophils to enhance their antifungal killing capacity during infection with Blastomyces dermatitidis (Bd). The study uses a well-rounded approach with multiple cell-type-specific SLAMF1 KO mice, in vivo and in vitro assays of fungal killing and assays of inter-cellular communications to arrive at the conclusions, which are well supported by the data. Overall, the manuscript is well written and provides an important advance in the field of antifungal immunity by identifying inter-cellular communication in mediating fungal clearance.

**Part II – Major Issues: Key Experiments Required for Acceptance**

Reviewer #1: 1. The presented data convincingly show the role of SLAMF1 in phagocytic fungal killing. Do the authors see any role for SLAMF1 in leukocyte accumulation/recruitment

2. The authors focus their attention on neutrophils as there were ~4-fold higher number of neutrophils associated with the fungus than monocytes. While this is reasonable, as monocytes also participate in fungal killing and harbor SLAMF1, does a similar homotypic interaction-mediated enhancement of killing is observed for monocytes too?

3. Does the absence of SLAMF1 on CD4 T cells lead to impaired host survival?

**Part III – Minor Issues: Editorial and Data Presentation Modifications**

Reviewer #1: 1. In Figure 1C, the labels hide the flow events on the plot. Please modify the label position so that the events are visible.

2. The mouse fungal challenge method should include additional details on route of administration, and how lungs were homogenized.

3. For the in vitro killing assay, please specify the effector-to-target ratios used for the ease of interpretation.

PLOS authors have the option to publish the peer review history of their article (what does this mean?). If published, this will include your full peer review and any attached files.

Reviewer #1: No

**Figure resubmission:**
---

## [Editor Report · Decision Letter 1]

19 May 2026

Dear Dr. Wuthrich,

We are pleased to inform you that your manuscript 'Homotypic interactions between SLAMF1 receptors on innate T cells and neutrophils regulate killing of fungi' has been provisionally accepted for publication in PLOS Pathogens.

Best regards,

Amariliz Rivera

Academic Editor

PLOS Pathogens

Michal Olszewski

Section Editor

PLOS Pathogens

Sumita Bhaduri-McIntosh

Editor-in-Chief

PLOS Pathogens

orcid.org/0000-0003-2946-9497

Michael Malim

Editor-in-Chief

PLOS Pathogens

orcid.org/0000-0002-7699-2064
---

## [Editor Report · Acceptance letter]

Dear Dr. Wüthrich,

We are delighted to inform you that your manuscript, "Homotypic interactions between SLAMF1 receptors on innate T cells and neutrophils regulate killing of fungi," has been formally accepted for publication in PLOS Pathogens.

Best regards,

Sumita Bhaduri-McIntosh

Editor-in-Chief

PLOS Pathogens

orcid.org/0000-0003-2946-9497

Michael Malim

Editor-in-Chief

PLOS Pathogens

orcid.org/0000-0002-7699-2064